# Modified Triple-Tuned Bandpass Filter with Two Concurrently Tuned Transmission Zeros

**DOI:** 10.3390/s22249760

**Published:** 2022-12-13

**Authors:** Mirosław Magnuski, Artur Noga, Maciej Surma, Dariusz Wójcik

**Affiliations:** Department of Electronics, Electrical Engineering and Microelectronics, Silesian University of Technology, Akademicka 16, 44-100 Gliwice, Poland

**Keywords:** tunable bandpass filter, constant fractional bandwidth, single control voltage, microstrip technology

## Abstract

In this paper, a modified triple-tuned microstrip bandpass filter is presented. The filter consists of inductively cross-coupled resonators tuned with varactors. The application of the additional source-load couplings together with resonator branch swapping results in two transmission zeros tuned concurrently with operating frequency. These transmission zeros placed on both sides of the passband significantly increase slope steepness in transition bands. The example filter tuned from 0.36 to 0.78 GHz and controlled by a single voltage was manufactured and validated by measurements. It has a constant fractional bandwidth of 11%, low in-band insertion loss ranging from 1.8 to 2.5 dB, and out-of-band attenuation up to 5 GHz without parasitic passbands. The obtained filter parameters made it useful for preselector networks.

## 1. Introduction

Due to the application of software-defined radio (SDR) technology, contemporary wireless systems can be easily used for the transmission and reception of a variety of data, starting with audio and video and ending with telemetric information from sensors. A common feature of all SDR receivers is the adoption of bandpass filters in their analog front-ends. Fixed-frequency filters or tunable filters are used to reduce interference coming out of the receiver operating channel and eliminate parasitic channels introduced by frequency conversion. For these reasons, preselecting filters is required to have adequate slope steepness in the transition bands and sufficient attenuation in the stop band [1]. Tunable preselectors for SDR receivers should additionally feature a small size together with low in-band attenuation, a wide tuning range, and a sufficiently high IP3 (third-order intercept point).

Practically, the varactor-tuned microstrip bandpass preselecting filters built of some number of coupled resonators are most often applied. Increasing the number of resonators improves the steepness of slopes in the transition bands at the expense of a higher in-band insertion loss (IL) [2,3]. For this reason, most planar filters presented in the literature are built of no more than four resonators [2,4,5,6,7,8,9,10,11,12,13]. It is observed that the in-band IL of the inductively coupled tuned microstrip bandpass filters decreases with increasing center frequency [2,5,7,9,10,14]. However, tuned filters with capacitive couplings behave in the opposite way, and their in-band IL increases with tuning frequency [15].

The steepness of the slopes could be improved by using additional transmission zeros (TZs) spaced in the transition bands of the filter response [2,4,5,6,7,8,9,10,12,13]. To improve the shape factor in tunable filters, the TZs should be allocated in both the lower and upper transition bands and tuned concurrently with the operating frequency. This effect could be achieved by applying multiple mutual couplings between the resonators of the filter. For these purposes, source-load coupling techniques [6,8], cross-coupling techniques [2,7,9,10] and mixed-coupling techniques [5] are often used. Similar effects are achievable when defected ground structure [16,17], open/short stubs [18], combining lowpass and bandpass filters [19] or combining stopband and bandpass techniques [20,21] are applied.

Tuned filters could have a constant absolute bandwidth (CAB) or constant fractional bandwidth (CFB). The CFB is obtained within the filters that adopt the coupling coefficient independent of frequency. Therefore, they can be tuned with a single control voltage [2,5,6,12,13]. The CAB is obtained by the variation of the coupling coefficients together with the tuning frequency. It could be achieved by applying the filter structure having coupling coefficients dependent on the frequency itself [22,23,24] or by applying additional varactors for continuous adjustment of the coupling coefficients. In the second case, two or more control voltages are applied [7,8,9,10,11,17].

A desired property of the preselecting filters is the high linearity represented by the IP3 value. A high IP3 value can be obtained more easily in filters having a small tuning range when the varactors operate in the linear region of their characteristic. For filters with a wide tuning range, the IP3 usually has a smaller value. Thus, obtaining a wide tuning range and adequate IP3 is a matter of compromise.

In our two previous works, double-tuned bandpass filters built of double-coupled resonators were presented [12,13]. Both filter networks were obtained by modification of the double-tuned filter prototypes. The modifications introduced additional couplings between the resonators. Due to extra couplings, the slope steepness of the proposed filters characteristics were improved by the appearance of a single [13] or a double TZ [12] located in the filter responses next to their passbands. This article describes a triple-tuned bandpass filter with inductive multiple couplings obtained by a similar modification. Additionally, increasing the number of resonators and using the source-load coupling technique improved the slope steepness in the transition bands. The flat frequency response in the passband with less in-band insertion loss and suppression of the first parasitic band were also achieved. A filter with two concurrently tuned transmission zeros working within the tuning range 0.36–0.78 GHz was designed, manufactured, and verified by measurements. The example filter has a constant fractional bandwidth of 11% and a shape factor of 1.98. Its insertion loss changes together with tuning from 1.7 to 2.5 dB and out-of-band attenuation decreases from 55 dB at 250 MHz to 25 dB at 1600 MHz. The obtained IP3 increases from 13.6d Bm at the lowest operating frequency to 32 dBm at the highest. The filter structure has small electrical dimensions of 0.08 × 0.09 $\lambda_{g}$, where $\lambda_{g}$ is the guided wavelength at the lowest operating frequency. The circuit and full-wave simulations of the filter were performed using CST Microwave Studio software.

This paper is organized as follows. In Section 2, the lumped elements prototype of the filter and its parametric analysis are described in detail. At the beginning of Section 3, the realization of the microstrip filter is shown considering the improvement of in-band return loss and out-of-band attenuation. The results of measurements of the example filter are then presented. In Section 4, the comparison of the designed filter with the other filters described in the literature is shown. Section 5 contains conclusions.

## 2. Lumped Element Prototype

The initial circuit for the designed bandpass filter is a triple-tuned filter with inductive couplings shown in Figure 1a. All resonators have the same center frequency. The filter network is extended by the addition of input series and output series inductances $L_{i}$ and $L_{o}$ interconnecting the input/output ports to the input/output resonators of the filter.

The first step of filter modification is shown in Figure 1b. Each of the coupling inductors $L_{c1}$ and $L_{c2}$ is split into two parts, and the ground connections are moved to the newly formed nodes. The division of both inductors is carried out with factor *k* (0<k<1) so that the first parts of the divided inductors have values of $kL_{c1}$, $kL_{c2}$ and the second parts have values of $\left(1-k\right)L_{c1}$, $\left(1-k\right)L_{c2}$. In the next step, the filter branches in the second and third resonators are swapped, as shown in Figure 1c.

The final structure is a triple-tuned multiple-coupled (TTMC) filter having each resonator coupled with the two others. The couplings within the filter are illustrated in Figure 2 by coloring the elements of each resonator. The elements of the first resonator are colored blue, the second orange, and the third green.

Since in multi-resonator filters, all resonators are tuned to the same frequency, the proposed filter must also meet this requirement. Therefore, assuming the equality of $L_{c1}$ and $L_{c2}$, the identity of the first and third resonators, and the equality of all capacitors, the inductance $L_{2}$ should satisfy the relationship
(1)$$L_{2}=L+L_{c}\left(2k-1\right).$$

According to this condition, the resonant frequency of each resonator is
(2)$$\omega_{0}=\frac{1}{C(L+L_{c})}.$$

The scattering parameters of the initial triple-tuned bandpass filter and the final TTMC filter calculated for C= 5.5 pF, $L_{i}=L_{o}=$ 3.63 nH, $L_{m}=$ 4.25 nH, $L_{c}=$ 1.4 nH, L= 13.1 nH, and k= 0.2 are shown in Figure 3 and Figure 4. Both bandpass filters have the same center frequency. However, because additional couplings are introduced between the input and output resonators of the TTMC filter, two TZs that shape the filter response in the transition bands are obtained. Therefore, the TTMC filter has much steeper slopes and a narrower bandwidth. The filter also has smaller ripples in the operating frequency band. Improvements in the passband and transition bands are achieved at the expense of decreasing out-of-band attenuation.

To illustrate the relationships between filter properties and element values, a parametric analysis was performed. As shown in Figure 5, an increase in the division factor *k* shifts the frequency of the left TZ upward, resulting in a narrower passband, increased slope steepness in the left transition band and for k≥0.2 improved impedance matching. The frequency of the right TZ and the right cutoff frequency are practically not changed, but the out-of-band attenuation worsens slightly.

Figure 6 shows the influence of the coupling inductance $L_{c}$. An increase in the $L_{c}$ value results in a widening of the passband, a slight decrease in the left TZ frequency, and a slight deterioration of the out-of-band attenuation. The characteristics of $s_{11}$ show the existence of an optimal value of $L_{c}$, which in the considered case is about 1.6 nH.

As can be seen in Figure 7, the inductance $L_{m}$ has a strong influence on the impedance matching of the filter, especially in the upper part of the passband. Increasing the value of $L_{m}$ increases the frequency of the right TZ. It also slightly affects the shape of the response within the passband (see Figure 7). For the filter under study, the optimal value of $L_{m}$ is about 4.25 nH.

Figure 8 shows the effect of a simultaneous change in the inductance of $L_{i}$ and $L_{o}$ on the response characteristics. An increase in out-of-band attenuation is observed as the values of both inductances increase. The $L_{i}$ and $L_{o}$ also have a strong impact on filter matching, especially at the lower part of the passband.

## 3. Tunable Microstrip Filter

It was assumed that the example filter is tuned in the frequency range from 400 to 800 MHz. In the HF (High Frequency) and VHF (Very High Frequency) bands, the filter can be constructed with lumped inductors. For UHF (Ultra High Frequency) and higher frequency bands, it is more convenient to realize the inductors as transmission line sections due to technological limitations caused by parasitic parameters of the inductors. The TTMC filter was realized as a tunable network with the use of microstrip transmission lines. Due to the necessity of applying transmission lines with high characteristic impedances, the circuit was made on a Rogers 5870 laminate with a low electrical permeability value $\varepsilon_{r}=$ 2.33 and a relatively high thickness of h= 1.5 mm.

Figure 9 shows a schematic diagram of the filter built of the transmission lines. The transmission lines and varactors of each resonator are colored as in Figure 2. All inductances within the network depicted in the schematic diagram shown in Figure 2 are replaced by the TL1-TL8 transmission lines, whose initial parameters were estimated by applying the following equation
(3)$$\begin{array}{ccc} \omega L & =Z_{0}\tan\beta l & \;\;\;\;\;\;\;\;\;\mathrm{for}\;f_{\max},\;\beta l\ll 1 \end{array}$$
describing the input reactance of the shorted transmission line. The assumed line widths, their characteristic impedances, and initial lengths determined from Equation (Equation 3) are summarized in Table 1.

Each capacitor is replaced by a push–pull connection of eight 1SV280 varactors with capacitance ranging from 1.2 to 5.5 pF, series resistance of 0.44 Ω and the lead inductance of 0.4 nH. The control voltage is delivered to the varactors through series resistors $R_{c}=$ 0.33 kΩ using filtering capacitors $C_{c}=$ 100 pF.

During the design of TL3, TL4 and TL6 lines that replace coupling inductors, the inductances of the vias are taken into account. To improve out-of-band attenuation, the characteristic impedances and physical lengths of the lines forming the resonators were differentiated while keeping the overall inductance of all resonators constant. The characteristic impedances of the lines TL1 replacing $L_{1}$ and $L_{3}$ are about 106 Ω, and the line TL5 replacing $L_{2}$ is about 160 Ω. Consequently, it results in the quality factors of the first and third resonators being above 100 and the second above 80 including varactor loss. For lossless varactors, the quality factors would be above 140 and 120, respectively.

Figure 10 shows $s_{21}$ in the frequency range up to 5 GHz, for filters with applied identical TL1 and TL5 lines or lines having different impedances and lengths. As can be seen, the applied differentiation of transmission lines does not change the filter response within its passband, but it improves the attenuation occurring for high frequencies.

To improve filter impedance matching in the upper part of the tuning range, the low-impedance sections of the TL8 line acting as 0.15 pF capacitors are added in series to the TL1 lines. The impact of these lines on the course of $s_{11}$ is illustrated in Figure 11.

The resulting filter was subjected to circuit and full-wave simulations. The final values of the line widths and lengths are summarized in Table 2. Figure 12 shows the final layout of the designed filter with the TL1–TL8 lines labeled. The TL8 lines are formed by enlarging the corresponding varactor pads within the first and third resonators. The layout also contains pads for the $R_{c}$ and $C_{c}$ components of the varactor polarization networks applied in each resonator. To reduce the dimensions of the entire layout, these elements were placed in the spare area inside the layout.

A photo of the assembled filter is shown in Figure 13. The total dimensions of the prototype, including the SMA (SubMiniature version A) connectors, are 52×42 mm. The filter parameters were measured with the Agilent PNA-L N5230A vector network analyzer. Figure 14 and Figure 15 show a comparison of the measured and simulated S parameters in the frequency range from 200 to 1000 MHz. During the measurements, the varactor polarization voltage varied from 1 to 25 V. For comparison purposes, the capacitance values set for the calculations were selected to achieve agreement of the center frequencies with those obtained by means of measurements. For legibility, the figures show the results for five selected center frequencies. As can be seen, very good agreement is achieved. The measured tuning range of the filter spreads from 360 to 780 MHz, instead of the assumed 400–800 MHz. The shift in the tuning range toward lower frequencies might be the result of the application of varactors having slightly higher capacitances than it was expected. In the tuning range, the filter bandwidth varies from 41 to 86.9 MHz.

The filter has a relatively low loss. The highest insertion loss of 2.5 dB was measured at the lowest center frequency of 360 MHz, and as the center frequency increases, it decreases to 1.9 dB at the maximum center frequency of 780 MHz. The attenuation for both TZs is slightly dependent on the center frequency and ranges from 50 to 60 dB. For frequencies below the first TZ, the IL exceeds 40 dB and above the second TZ, it is better than 30 dB up to 1 GHz. The $s_{11}$ in the passband varies with the change in the filter center frequency and is less than −9 dB in the entire tuning range. The best impedance matching is achieved for the sub-band from 450 to 600 MHz. However, the best agreement between measurements and simulations is observed at both ends of the tuning range.

Figure 16 shows the results of the measurements and simulations of $s_{21}$ in the range up to 5 GHz for the three different center frequencies. It is visual that the level of out-of-band attenuation practically does not depend on the varactors control voltage. The smallest attenuation level of about 12 dB occurs for 2375 MHz. There are no parasitic passbands up to 5 GHz.

To illustrate the reason for the appearance of the additional TZs in the filter response, current density distributions were determined for the selected center frequency of 585 MHz and the TZ frequencies of 490 MHz and 700 MHz. The results of the analyses are shown in Figure 17, Figure 18 and Figure 19, respectively. For a center frequency of 585 MHz, the current density distributions are similar in all resonators, and the signal is transferred between the input and output ports with little loss. For both TZ frequencies (490 MHz and 700 MHz), the current density distributions in the resonators are different. The compensation of currents flowing to the output takes place (currents $I_{1}$, $I_{2}$ and $I_{3}$ in Figure 17, Figure 18 and Figure 19), which is the reason for the high signal attenuation of 80 dB. This effect is also shown in Figure 20 and Figure 21, where the amplitudes of the currents $I_{1}$ and $I_{2}$ at the TZ frequencies are the same and their phases are opposite.

IP3 of the prototype filter was measured with two HP 8665B signal generators, a directional combiner, and an R&S FSV30 spectrum analyzer. The measurement was performed with a two-tone offset of 1 MHz and a signal level of −5 dBm at the combiner output. The results are presented in Figure 22. The IP3 varies from 13.5 to 32.5 dBm within the tuning range, which seems to be sufficient for most input stages applied in receivers.

## 4. Discussion

Table 3 compares the triple-tuned multiple-coupled filter with similar bandpass filters published in the literature. The proposed TTMC filter is built with the three mutually coupled resonators. It is tuned by means of varactors with a single control voltage. The filter has a constant fractional bandwidth of 11% within the tuning range 0.36–0.78 GHz ($f_{\max}/f_{\min}=$ 2.17). Due to the TZs located in the transition bands, the proposed filter has a shape factor (SF) equal to 1.98. The SF is defined as
(4)$$\mathrm{SF}=\frac{{\mathrm{BW}}_{20}}{{\mathrm{BW}}_{3}},$$
where ${\mathrm{BW}}_{20}$ and ${\mathrm{BW}}_{3}$ are 20 dB and 3 dB bandwidths, respectively.

The filter insertion loss changes with the tuning frequency from 1.8 to 2.5 dB. The out-of-band attenuation decreases from 55 dB at 250 MHz to 25 dB at 1600 MHz. The IP3 depends on the varactors operating point, and it increases from 13.6 dBm to 32 dBm together with the increase of the varactor control voltage from 1 to 25 V. The filter has electrical dimensions of 0.08 × 0.09 $\lambda_{g}$, where $\lambda_{g}$ is the guided wavelength at the lowest operating frequency.

A filter having the largest tuning range is found in [6]. Although the design consists of two resonators, it has a larger electrical size and a higher maximum IL value than the proposed. All the compared filters have the bandwidth of 5% to 15%. The best shape factor of 1.55 has the filter [2] built of four resonators, but it occurs for a narrow tuning range of 22%. The proposed filter has the best shape factor of 1.98 among all compared filters built of two or three resonators. Most of the filters compared in Table 3 have two TZs. The filters [5,11] have four and three TZs, respectively. The presence of four TZs in the frequency response allows the filter [5] to achieve the greatest shape factor of 1.8. The smallest IL variation from 0.82 to 2.03 dB is achieved by the triple-tuned filter built of concentric resonators [8]. However, the proposed filter has the smallest IL among all the compared microstrip designs. The second in this respect is a double-tuned filter which has a variation of IL from 1.9 to 3.4 dB [12]. The highest IP3, greater than 30 dBm, has the filter [4]. This is achieved due to the operation in the more linear region of the varactor tuning characteristics, and it results in a narrower tuning range of 1.57. The presented TTMC filter has the third smallest electrical size of all compared filters. The smaller are two filters designed by the authors: a double-tuned and a quadruple-tuned [12].

## 5. Conclusions

This paper describes the constant fractional bandwidth triple tuned microstrip bandpass filter. Thanks to the application of the source-load couplings together with resonator branch swapping, two transmission zeros tuned concurrently with operating frequency are achieved. The TZs are located on both sides of the passband, which significantly increases the shape factor of the filter. The example filter is tuned from 0.36 to 0.78 GHz and is controlled by a single voltage. It has a constant fractional bandwidth of 11% and low in-band insertion loss ranging from 1.8 to 2.5 dB. Its out-of-band attenuation decreases from 55 dB at 250 MHz to 25 dB at 1600 MHz. The obtained IP3 increases from 13.6 dBm at the lowest operating frequency to 32 dBm at the highest. The filter structure has small electrical dimensions of 0.08 × 0.09 $\lambda_{g}$. The obtained filter parameters made it useful for preselector networks. Future research will focus on the filter structures with a constant absolute bandwidth.

## Figures and Tables

**Figure 1 sensors-22-09760-f001:**
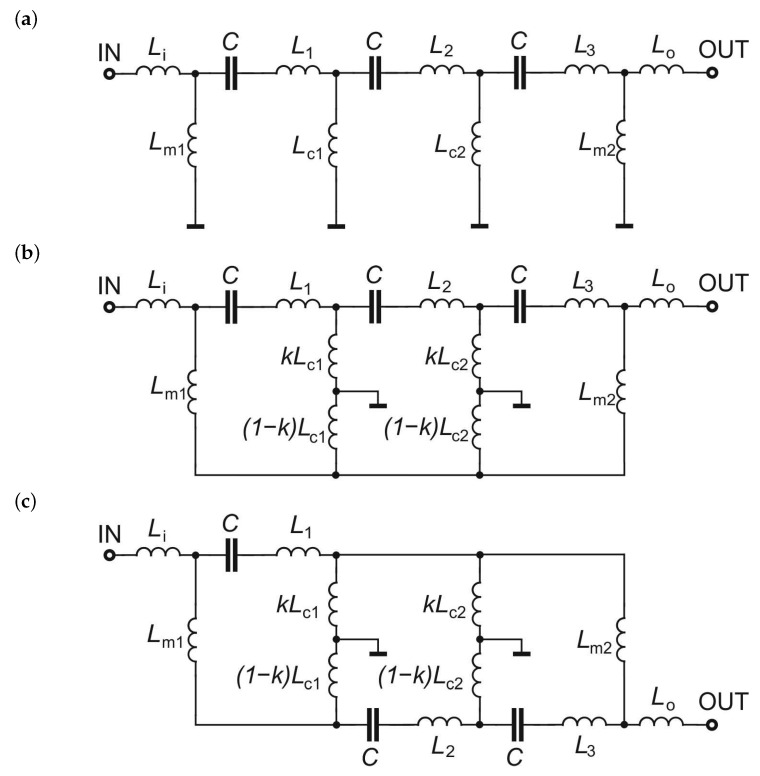
Transforming of the lumped element bandpass filter: (**a**) the initial filter, (**b**) the filter with split coupling inductances, and (**c**) the final TTMC filter with swapped branches.

**Figure 2 sensors-22-09760-f002:**
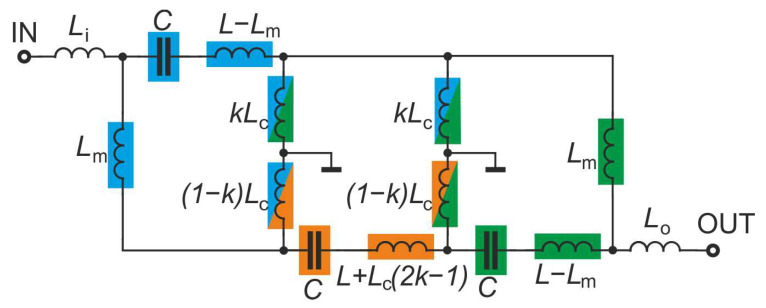
Lumped element triple-tuned multiple-coupled bandpass filter.

**Figure 3 sensors-22-09760-f003:**
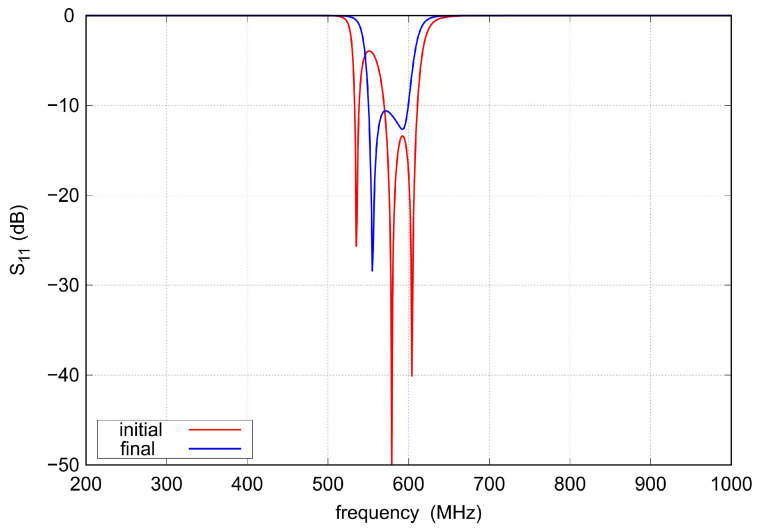
The $s_{11}$ parameter of the lumped element bandpass filters shown in Figure 1 (k= 0.2, C= 5.5 pF).

**Figure 4 sensors-22-09760-f004:**
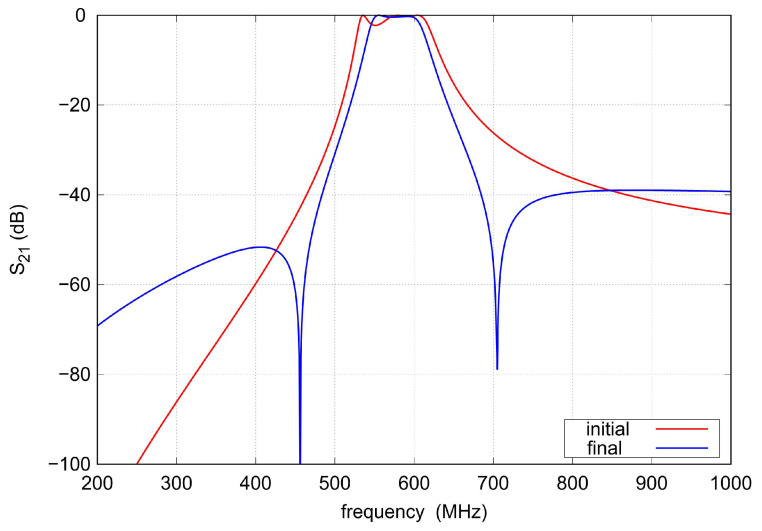
The $s_{21}$ parameter of the lumped element bandpass filters shown in Figure 1 (k= 0.2, C= 5.5 pF).

**Figure 5 sensors-22-09760-f005:**
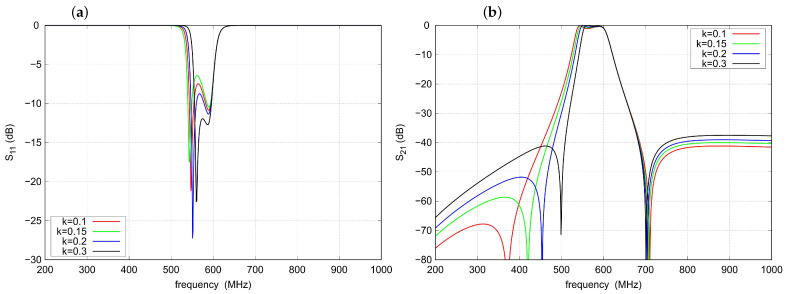
Effect of the division coefficient *k* on the filter response: (**a**) $s_{11}$ parameter, (**b**) $s_{21}$ parameter.

**Figure 6 sensors-22-09760-f006:**
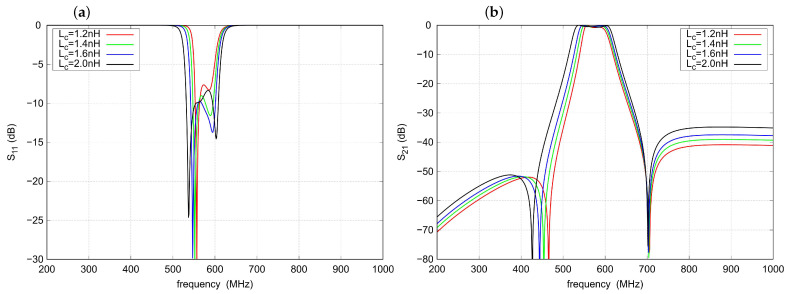
Influence of the $L_{c}$ inductance on the filter response: (**a**) $s_{11}$ parameter, (**b**) $s_{21}$ parameter.

**Figure 7 sensors-22-09760-f007:**
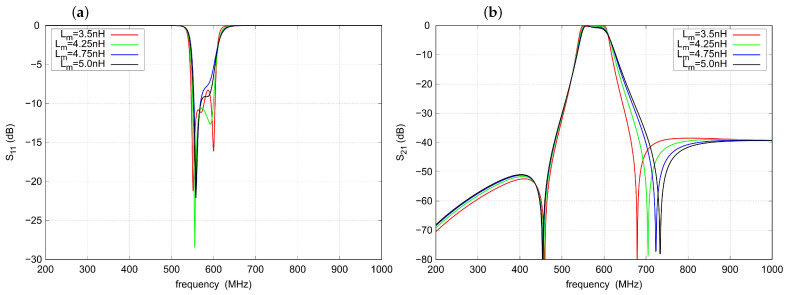
Influence of the $L_{m}$ inductance on the filter response: (**a**) $s_{11}$ parameter, (**b**) $s_{21}$ parameter.

**Figure 8 sensors-22-09760-f008:**
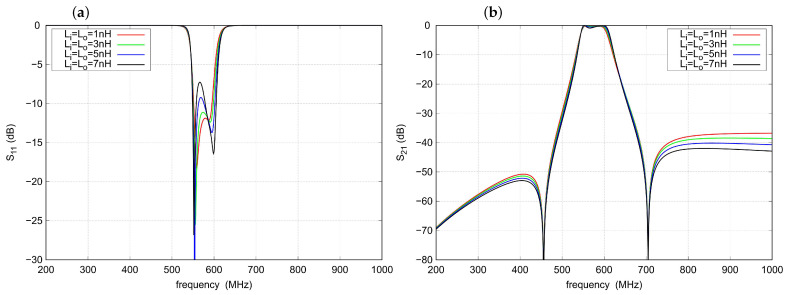
Influence of the $L_{i}$ and $L_{o}$ inductances on the filter response: (**a**) $s_{11}$ parameter, (**b**) $s_{21}$ parameter.

**Figure 9 sensors-22-09760-f009:**
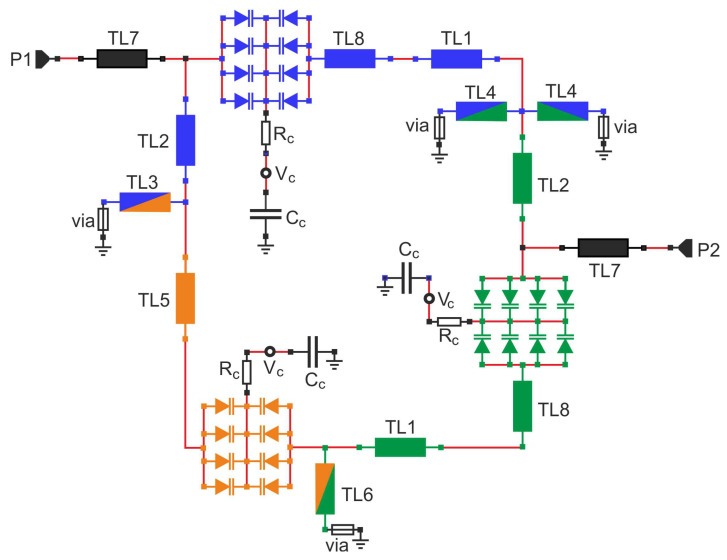
Transmission line implementation of the tunable TTMC filter.

**Figure 10 sensors-22-09760-f010:**
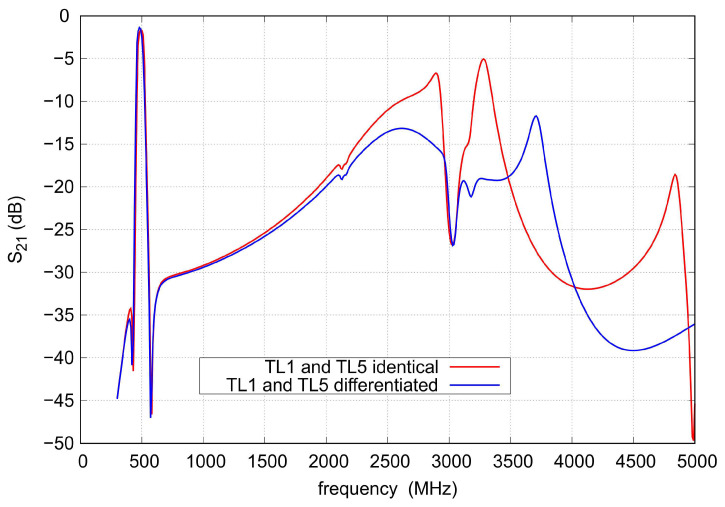
The $S_{21}$ for different impedances of transmission lines forming resonant circuits.

**Figure 11 sensors-22-09760-f011:**
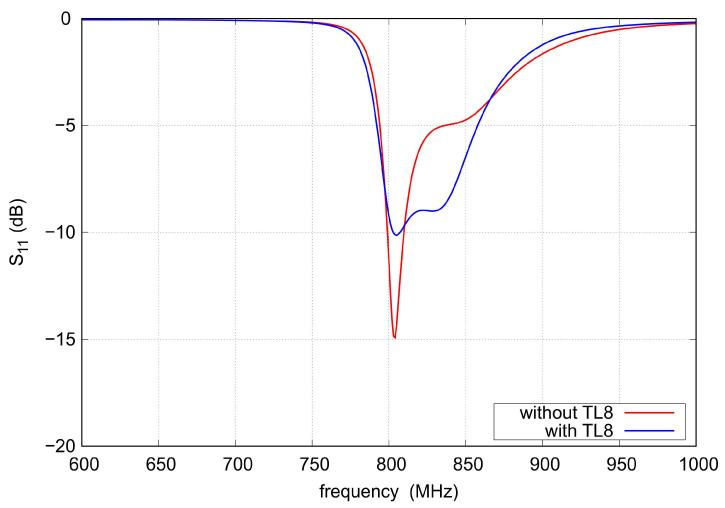
The $S_{11}$ calculated for circuit with and without TL8.

**Figure 12 sensors-22-09760-f012:**
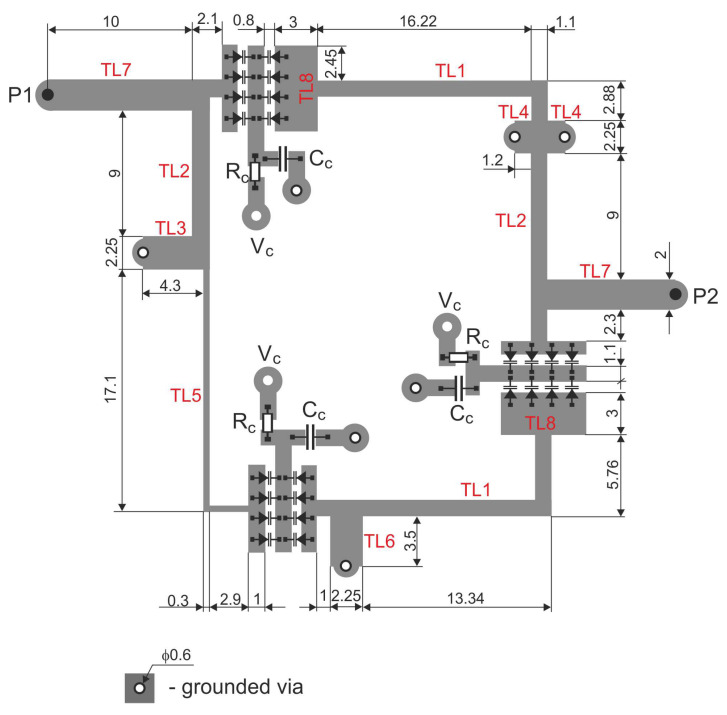
Layout of the proposed tunable bandpass filter (all dimensions in mm).

**Figure 13 sensors-22-09760-f013:**
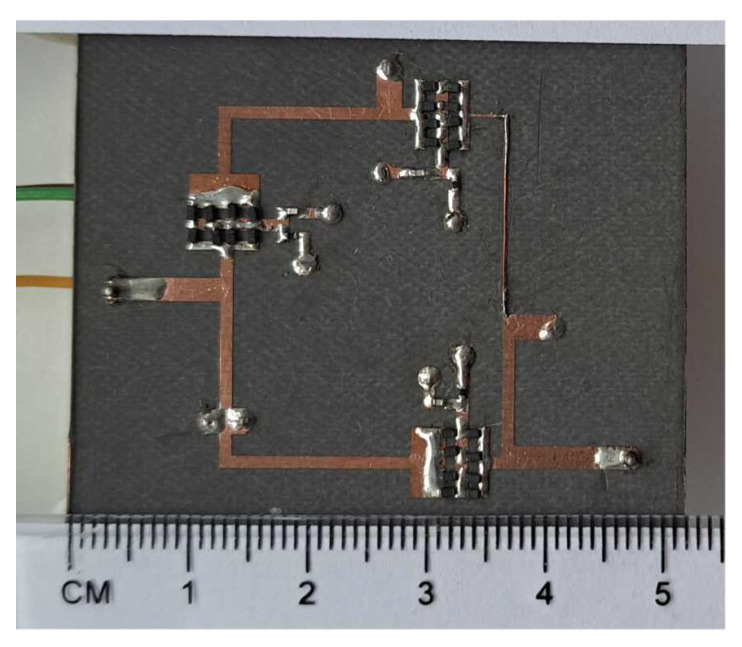
Photo of the fabricated prototype of the triple-tuned filter.

**Figure 14 sensors-22-09760-f014:**
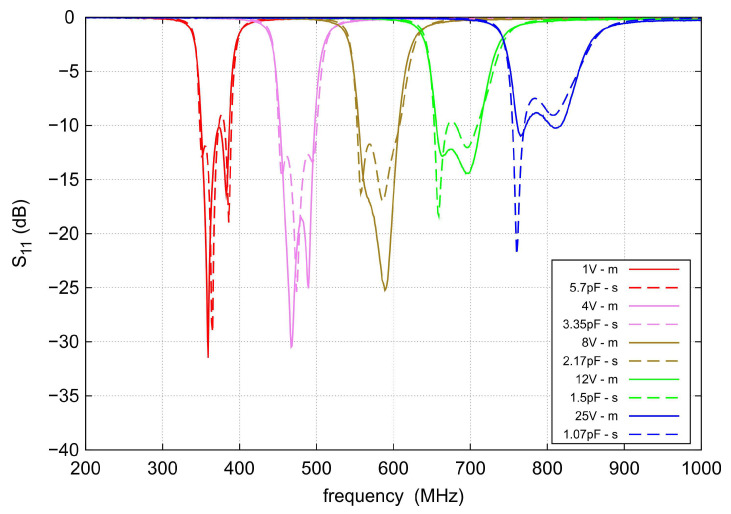
Measured (solid lines) and simulated (dashed lines) $s_{11}$ of the proposed tunable bandpass filter for the five selected control voltages.

**Figure 15 sensors-22-09760-f015:**
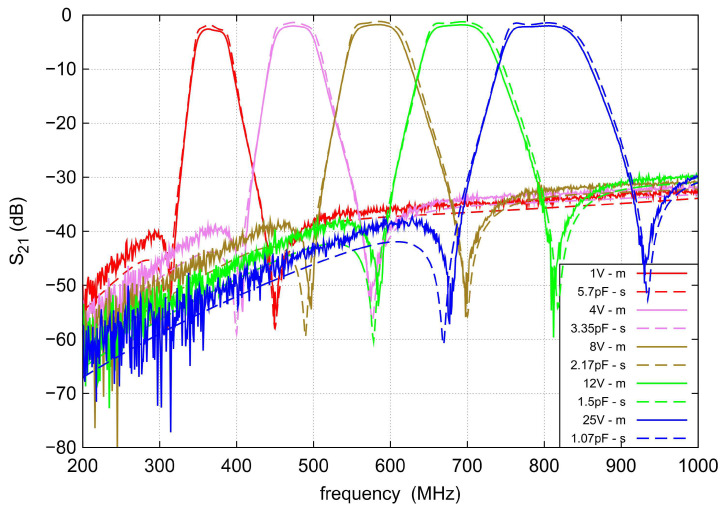
Measured (solid lines) and simulated (dashed lines) $s_{21}$ of the proposed tunable bandpass filter for the five selected control voltages.

**Figure 16 sensors-22-09760-f016:**
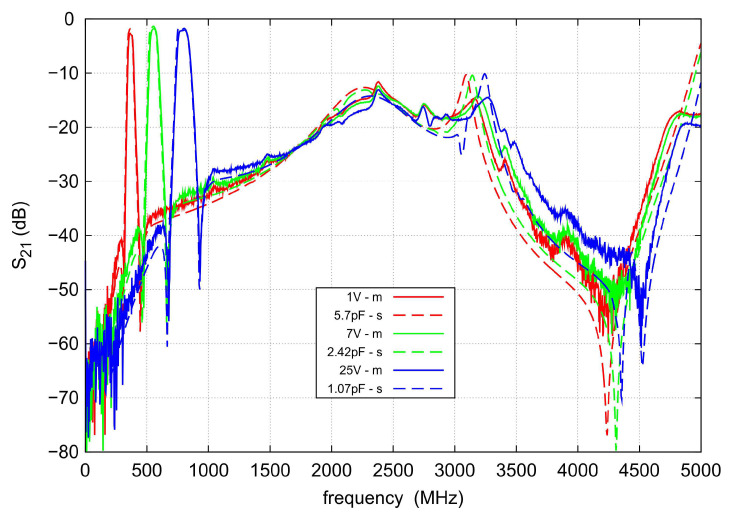
Broadband measurements (solid lines) and simulations (dashed lines) of the $s_{21}$ for the three selected control voltages.

**Figure 17 sensors-22-09760-f017:**
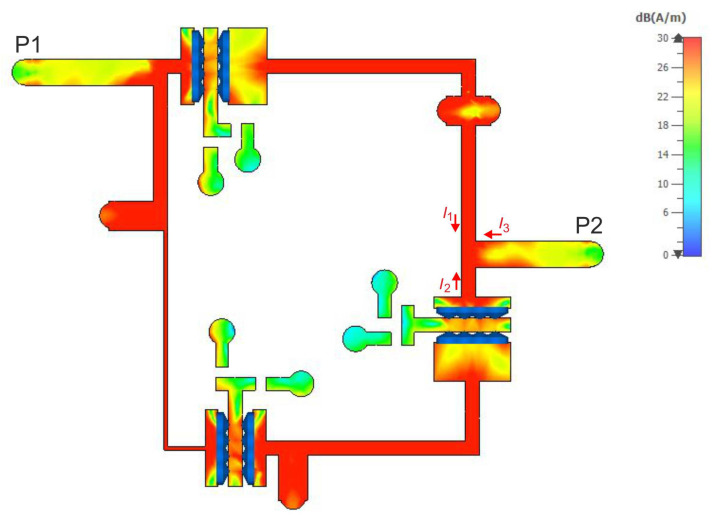
Current density distribution at the center frequency (585 MHz) of the filter.

**Figure 18 sensors-22-09760-f018:**
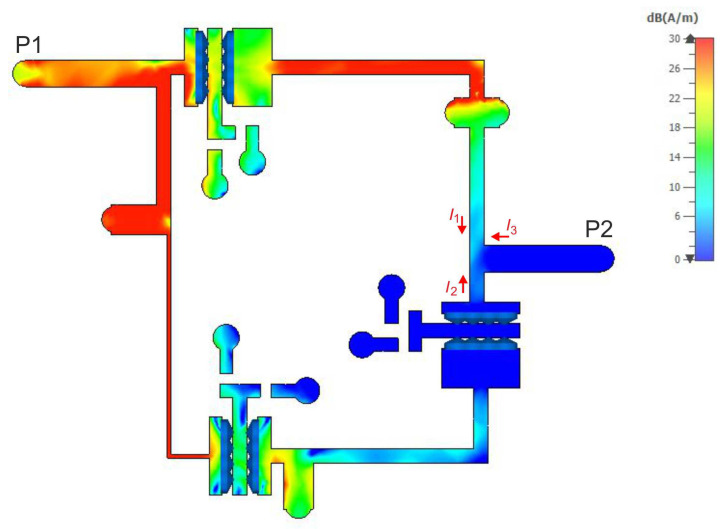
Current density distribution at the lower TZ frequency (490 MHz).

**Figure 19 sensors-22-09760-f019:**
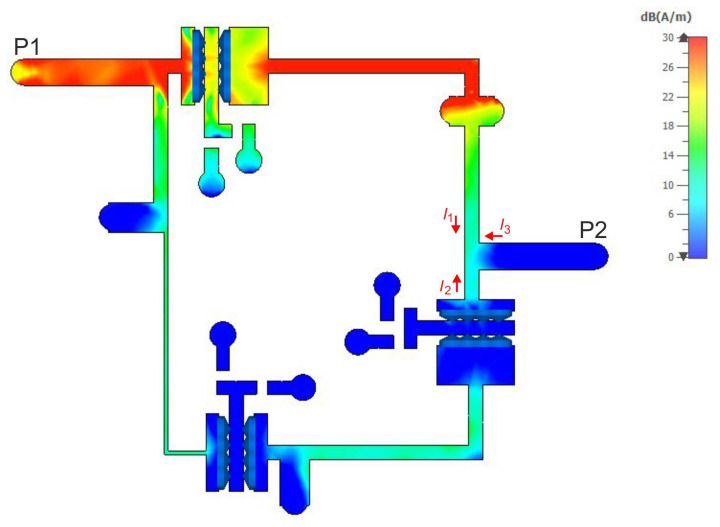
Current density distribution at the upper TZ frequency (700 MHz).

**Figure 20 sensors-22-09760-f020:**
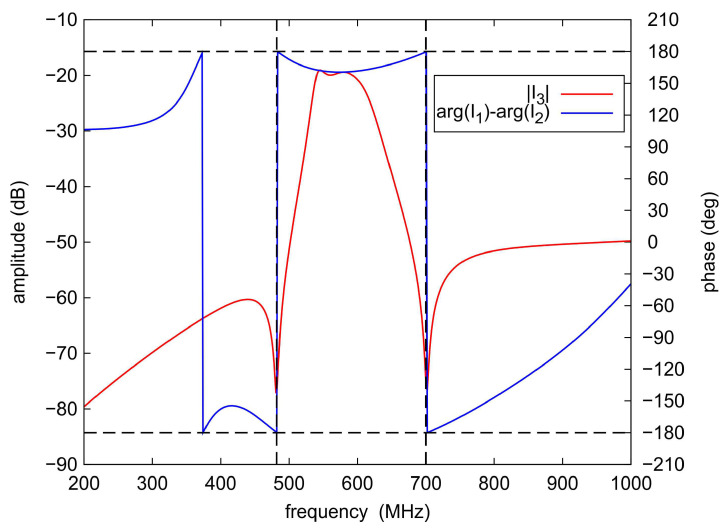
The I3 current amplitude and the phase difference of the $I_{1}$ and $I_{2}$ currents at the output node.

**Figure 21 sensors-22-09760-f021:**
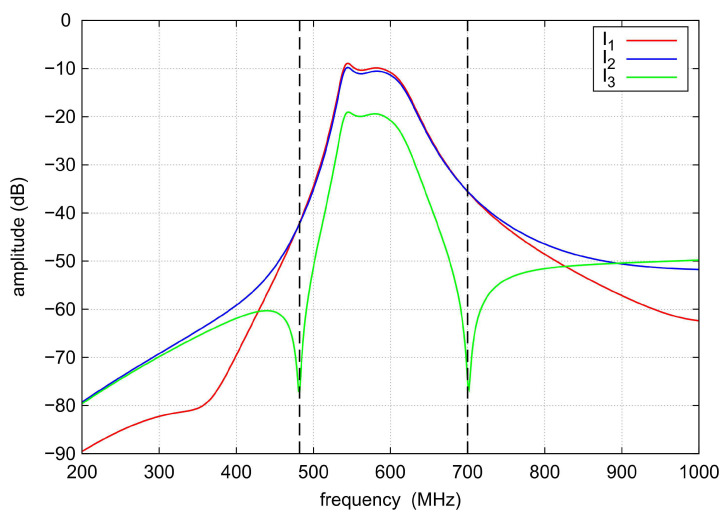
Amplitudes of the currents flowing through the output node.

**Figure 22 sensors-22-09760-f022:**
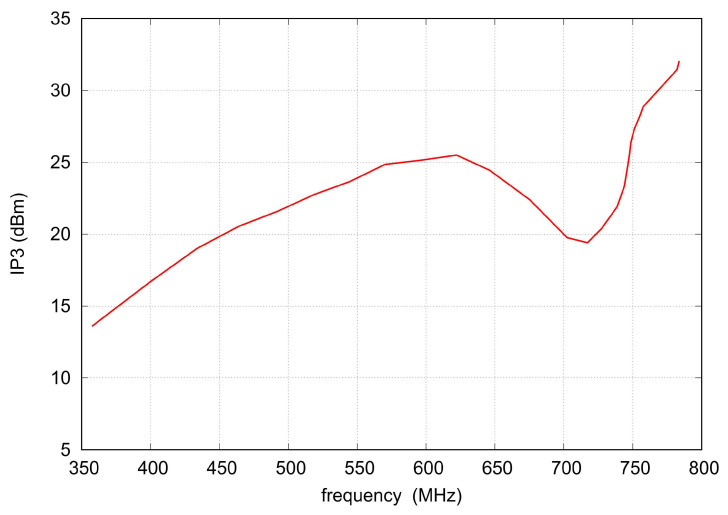
Measured IP3 versus filter center frequency.

**Table 1 sensors-22-09760-t001:** Initial values of transmission lines of the filter from Figure 9.

Line	Width (mm)	$Z_{0}$ (Ω)	βl (deg) ${}^{*}$
TL1	1.09	106	22.3
TL2	1.09	106	12.3
TL3	2.24	75	4.6
TL4	2.24	75	1.6
TL5	0.3	160	23.4
TL6	2.24	75	4.6
TL7	2.05	79	13.2
TL8	8	33	5.5

*—calculated at 800 MHz.

**Table 2 sensors-22-09760-t002:** Final values of transmission lines of the filter from Figure 9 and Figure 12.

Line	w (mm)	$Z_{0}$ (Ω)	βl (deg) ${}^{*}$
TL1	1.1	105	24.8
TL2	1.1	105	11.7
TL3	2.25	75	4.6
TL4	2.25	75	1.6
TL5	0.3	160	25.4
TL6	2.25	75	4.6
TL7	2.0	80	13.2
TL8	6	41	4.1

*—calculated at 800 MHz.

**Table 3 sensors-22-09760-t003:** Comparison between the proposed and the reference filters.

Ref.	f (GHz)	$f_{\max}/f_{\min}$	BW (MHz)	IL (dB)	Size ^1^ ($\lambda_{g}\times \lambda_{g}$)	NTZs	NV	IP3 (dBm)	SF	NR
[2]	2–2.45	1.22	5.19–6.75%	3.15–6.85	0.4 × 0.4	2	1	4.5–28	1.55	4
[4]	1.6–2.27	1.41	137 ± 2 6–8.5%	1.99–4.17	0.5 × 0.4	2	1	30–33	2.73	2
[5]	0.97–1.53	1.57	5.5 %	2–4.2	0.09 × 0.1	4	1	8–30.3	1.8	4
[6]	0.75–1.87	2.49	75–285 10–15.2%	1.2–4.2	0.38 × 0.13	2	1	15.5–25.5	2.26	2
[7]	0.95–1.48	1.55	114–120	3.5–4.4	–	2	2	25.2–34.9	1.75	4
[8]	0.7–0.96	1.37	35.2 ± 2.7	0.82–2.03	0.125 × 0.15 × 0.1	2	2	7.8–28	2.33	3
[9]	1.75–2.25	1.28	70–110	5.7–6.7	0.2 × 0.28	2	2	10.5–22.5	2	3
[10]	1.47–1.83	1.24	108 ± 2 5.9–7.3%	3.71–4.19	0.54 × 0.33	–	2	–	2.15	3
[11]	1.5–2.5	1.66	162	–	0.17 × 0.2	3	2	–	4.8	3
[12] ^2^	0.37–0.8	2.16	23–48 (6%)	1.9–3.4	0.05 × 0.06	2	1	8–18	3.15	2
[12] ^3^	0.38–0.79	2.08	17.5–35.5 (4.6%)	3.5–6.9	0.075 × 0.1	2	1	–	1.99	4
This work	0.36–0.78	2.17	41–87 (11%)	1.8–2.5	0.08 × 0.09	2	1	13.6–32	1.98	3

^1^ Calculated for the lowest operating frequency (given in the second column). ^2^ Single section filter. ^3^ Twosection filter.

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
