# Peer review of "Modified Triple-Tuned Bandpass Filter with Two Concurrently Tuned Transmission Zeros"

_sensors, 2022, doi:10.3390/s22249760_

Round 1

Reviewer 1 Report

Reviewer Comments

Manuscript Title: Modified triple-tuned bandpass filter with two concurrently tuned transmission zeros

Manuscript Number: sensors-2072121

The manuscript under review is devoted to simulating and studying a modified triple-tuned microstrip bandpass filter. The filter consists of inductively cross-coupled resonators tuned with varactors. Authors apply additional source-load couplings together with resonator branch swapping results in two transmission zeros tuned concurrently with operating frequency which permits to increase the slope steepness in transition bands. The proposed filter tuned from 0.36 5

to 0.78 GHz and controlled by a single voltage was manufactured and validated by measurements.

The manuscript contains new and significant. The abstract clearly and accurately describes the content of the article. The literature review part contains distinct and rich references. The paper is nicely written and can get accepted but first, it should be improved. I have these comments:

1- Authors define the majority of abbreviation through paragraphs and at the end of the manuscript, but some abbreviation still not defined for example IP3 in line 22. Please verify all abreviation.

2- It will be better if the authors describe the connection between paragraphs 2 and 3. What is the relation between Figure 2 and figure 9.

3- Line 133, authors announced that all inductances are replaced by the TL1-TL8 transmission lines. Why ? is not possible to simulate and fabricate the filter with inductances? What are the advantages of this replacement?

Reviewer 2 Report

The paper describes the constant fractional bandwidth triple-tuned microstrip band-pass filter with two transmission zeros in the response. The wide out-of-band rejection and good insertion losses are achieved. The reviewer suggests a thorough proofread of the manuscript by an English expert.

Reviewer 3 Report

In this paper, a modified triple-tuned microstrip bandpass filter which consists of inductively cross-coupled resonators tuned with varactors is presented. Here are some comments and suggestions.

1. Can the TZs and center frequency of the filter be tuned independently?

2. Please calculate the Q-factor of the filter and analyze the influence after adding the varactors.
